# Cardiac Damage and Conduction Disorders after Transcatheter Aortic Valve Implantation

**DOI:** 10.3390/jcm13020409

**Published:** 2024-01-11

**Authors:** François Damas, Mai-Linh Nguyen Trung, Adriana Postolache, Hélène Petitjean, Mathieu Lempereur, Tommaso Viva, Cécile Oury, Raluca Dulgheru, Patrizio Lancellotti

**Affiliations:** 1Department of Cardiology, CHU Sart Tilman, GIGA Cardiovascular Sciences Liège, University of Liège Hospital, 4000 Liège, Belgium; fdamas@chuliege.be (F.D.); mlnguyentrung@chuliege.be (M.-L.N.T.); adriana.postolache@chuliege.be (A.P.); hpetitjean@chuliege.be (H.P.); mathieu.lempereur@chuliege.be (M.L.); tommaso.viva@unimi.it (T.V.); cecile.oury@uliege.be (C.O.); redulgheru@chuliege.be (R.D.); 2Department of Minimally Invasive Cardiac Surgery, University of Milan, 20122 Milan, Italy; 3IRCCS Galeazzi, Sant’Ambrogio Hospital, 20157 Milan, Italy

**Keywords:** TAVI, cardiac damage, conduction disorders, risk stratification

## Abstract

Recently, a staging system using 4 grades has been proposed to quantify the extent of cardiac damage associated with aortic stenosis (AS), namely AS-related cardiac damage staging (ASCDS). ASCDS is independently associated with all-cause mortality and important clinical outcomes. To evaluate whether it might be associated with the occurrence of conduction system disorders after TAVI, a total of 119 symptomatic patients with severe AS who underwent a TAVI were categorized according to ASCDS: group 1 (13.5%): no or LV damage; group 2 (58.8%): left atrial/mitral valve damage, atrial fibrillation (AF); group 3 (27.7%): low-flow state, pulmonary vasculature/tricuspid valve/RV damage. After TAVI, 34% of patients exhibited LBBB and 10% high-degree atrioventricular block (HD-AVB). No patient in group 1 developed HD-AVB whereas new LBBB was frequent in groups 2 and 3. Twenty-one patients presented with paroxysmal AF with a higher rate for each group increment (group 1: *n* = 0, 0%; group 2: *n* = 11, 15.7%; group 3: *n* = 10, 30.3%) (*p* = 0.012). Patients in group 3 had the higher rate of permanent pacemaker implantation (PPMI) (group 1: *n* = 1, 6.3%; group 2: *n* = 7, 10%; group 3: *n* = 9, 27.3%) (*p* = 0.012). In conclusion, ASCDS might help identify patients at higher risk of conduction disorders and PPMI requirement after TAVI.

## 1. Introduction

Transcatheter aortic valve implantation (TAVI) is now the standard of care for patients with severe symptomatic aortic stenosis (AS) and has been validated in all surgical risk categories [1]. In the European guidelines, TAVI should be the preferred treatment option for patients aged ≥75 years with accessible transfemoral approach or for those at high surgical risk (STS-PROM/EuroSCORE II > 8%) [2]. Nevertheless, although TAVI is less invasive than surgical aortic valve replacement (SAVR), patients require demanding care monitoring. TAVI-related conduction disturbances, mainly new-onset left bundle-branch block (LBBB) and high-degree atrioventricular block (HD-AVB) necessitating permanent pacemaker implantation (PPMI), are still the most common complications encountered after this procedure and may have a significant detrimental association with patients prognosis [3,4,5,6]. The anatomical contiguity between the aortic valve and the conduction system underlies the genesis of periprocedural conduction disturbances during TAVI. Numerous factors have been associated with an increased risk of conduction disturbances and PPMI. Prior right bundle-branch block (RBBB) and transcatheter valve type and implantation depth are the most frequently observed [7,8,9,10]. However, in addition to the mechanical interplay involving the prosthetic valve and the conduction system, substantial evidence buttresses the association between AS and conduction disturbances. Calcium deposition on the conduction system and left ventricular (LV) dysfunction have been both linked with the occurrence of LBBB and HD-AVB in patients with AS [11,12,13]. Delayed intervention in AS may result in irreversible myocardial damage that may lead to an increase in peri-procedural risks and adversely affect long-term clinical outcomes. Recently, a staging system using 4 grades has been suggested to assess the extent of cardiac damage related to AS, namely AS-related cardiac damage staging (ASCDS), which encompasses more than the aortic valve and the left ventricle (LV), extending to the left atrium, pulmonary circulation, and to the right ventricle (RV) and right atrium. Progressive stages of cardiac damage are independently associated with all-cause mortality and important clinical outcomes, such as post-TAVI readmission rates [14,15]. In the present study, we hypothesised that the extent of cardiac damage might be associated with the occurrence of conduction system disorders after TAVI.

## 2. Methods

### 2.1. Study Population

The study population consisted of 153 consecutive patients with symptomatic severe AS who underwent a TAVI procedure with a self-expandable valve (CoreValve EvolutR^®^, Medtronic Inc., Minneapolis, MN, USA) between January 2018 and December 2020 at the University of Liège Hospital. Data were collected from the patient hospital records or via the local health network (Walloon health network). The decision-making process for TAVI involved consensus among the institutional heart team. Patients with pre-existing intracardiac devices (PPM), with moderate to severe mitral stenosis or undergoing a valve in valve procedure were excluded (n = 34). A total of 119 patients were finally included. The study was conducted in accordance with the Declaration of Helsinki and approved by the Ethics Committee of Liege University Hospital (protocol code: 2021/306, date of approval: 12 October 2021).

### 2.2. Echocardiography and Cardiac Staging

Conducted by an experienced cardiologist adept in valvular heart disease assessment, transthoracic echocardiography using a Vivid 95 GE machine was followed by analysis of echocardiographic images through EchoPac software v204 (GE Vingmed Ultrasound AS, Horten, Norway). A pre-TAVI procedure TTE was performed in all patients. The presence of severe AS, chambers dimensions and volumes, LV diastolic function and filling pressures, LV mass, LV and RV function, pulmonary pressure estimation and valvular regurgitation evaluation were defined according to current guidelines. The biplane method of discs’ summation (modified Simpson’s rule) was applied to quantify LV end-diastolic and end-systolic volumes and ejection fraction. The degree of AS was evaluated according to standard methods. LV stroke volume was calculated using the Doppler (LV outflow tract area × LV outflow tract velocity–time integral measured by pulsed-wave Doppler) method. The LV diastolic function was evaluated by the analysis of the mitral inflow velocities (E and A waves) and by Doppler tissue imaging measurement of early diastolic mitral annular velocities (e’). Systolic pulmonary arterial pressure (sPAP) was derived from the regurgitant jet of tricuspid regurgitation using systolic transtricuspid pressure gradient and the addition of 10 mmHg for right atrial pressure as previously performed. Left atrial maximal volume was measured using the modified Simpson’s rule. For each measurement, at least two cardiac cycles were averaged. Patients were categorized into five stages according to the extent of ASCDS: stage 0: no cardiac damage; stage 1, LV damage: LV hypertrophy (LV mass index > 95 g/m^2^ for women, >115 g/m^2^ for men), and/or LV diastolic dysfunction ≥ grade 2 and/or LV systolic dysfunction (LV ejection fraction < 60%); stage 2, left atrial and/or mitral valve damage: left atrial enlargement (left atrium volume index > 34 mL/m^2^) and/or ≥moderate mitral regurgitation, and/or AF; stage 3, pulmonary vasculature or tricuspid valve damage: PH defined as sPAP ≥ 60 mmHg, and/or ≥moderate tricuspid regurgitation; stage 4, RV damage and/or subclinical heart failure: RV dysfunction based on a multiparametric evaluation (TAPSE < 17 mm and s’ < 9.5 cm/s and fractional area change < 35%) and/or low-flow state (stroke volume index < 30 mL/m^2^). Patients were hierarchically classified in a given stage (worst stage) if at least one of the proposed criteria was satisfied. Given the small number of patients observed in stages 0–1 and 4, we categorized our population in 3 groups: group 1 including stages 0 and 1, group 2 including stage 2, group 3 including stages 3–4 [14,15,16].

### 2.3. TAVI Procedure and ECG Assessment

Most TAVI procedures were completed through transfemoral access under locoregional anaesthesia combined with sedation. In some patients, the access was subclavian or carotid artery with some cases requiring surgical approach to the vessel. The selection of valve size (23 to 34 mm) was made by the heart team based on preoperative cardiac computed tomography (CT) analysis. At the end of the procedure, the evaluation of the result was based on the correct positioning of the prosthesis and the absence of complications. All possible per-procedural complications have been described in the interventional cardiologist’s report: AF, high-grade conduction disorders, neurological problems suggesting ischaemic vascular accident, or death. In the event of insufficient deployment of the prosthesis and/or significant insufficiency, additional balloon dilatation was performed. Continuous ECG monitoring was implemented for patients during the initial 24 h following TAVI. If notable conduction irregularities were identified, this monitoring duration was prolonged. Subsequently, patients underwent daily ECG evaluations, even on the day of discharge. These assessments aimed to detect and track any new conduction irregularities that persisted until PPMI or discharge. New onset LBBB and RBBB were determined according to standard criteria. LBBB was defined as a QRS duration ≥ 120 ms with broad notched or slurred R wave in leads I, aVL, V5 and V6 and occasional RS pattern in V5 and V6 attributed to displaced transitions of QRS complex, absence of q waves in leads I, aVL, V5 and V6 (a narrow q wave can be present), R wave with slow growth in V1 to V3 with possible occurrence of QS, widened S waves with thickening and/or slots in V1 and V2, intrinsicoid deflection in V5 and V6 ≥ 55 ms, electrical axis between −30° and +60°; ST depression and asymmetrical T wave in opposition to medium-terminal delay. The hallmark of RBBB was a QRS duration ≥ 120 ms, larger R’-wave in V1/V2 and a broad and deep S-wave in V5/V6. HD-AVB included second-degree type 2 (BAV 2 type 2) and third-degree (BAV 3) atrioventricular blocks. Patients who necessitated PPMI underwent device implantation typically within the initial 3 days following TAVI, while still hospitalized. The decision for implantation involved collaborative input from the structural intervention and electrophysiology teams, in consultation with the patient.

### 2.4. Statistical Analysis

Qualitative variables are presented as count and percentage and comparisons were performed by Chi-square test or Fisher exact test when appropriate. Quantitative variables are presented as mean ± SD and comparisons were performed by ANOVA test. The evolution of electrocardiographic parameters was analysed by the paired Wilcoxon test. Logistic regression analysis was used to determine the co-factors associated with periprocedural PPMI. Odds ratios (OR) are calculated with their 95% confidence intervals. All tests are two-tailed and a *p*-value < 0.05 is considered significant. All statistical analyses were performed in SAS 9.4 (SAS Institute, Cary, NC, USA).

## 3. Results

### 3.1. Baseline Characteristics of the Study Population

Baseline pre-TAVI data according to the stage of cardiac involvement (group 1: 13.5%; group 2: 58.8%; group 3:27.7%) are summarized in Table 1. Overall, the general characteristics, except for age (*p* = 0.017) and the presence of AF (*p* = 0.003), were similar between the groups. Patients in group 1 were younger (78 years median age) compared to those in groups 2 and 3 (median age: 78 vs. 83.5 vs. 85 years; *p* = 0.046, *p* = 0.009, respectively). The rate of AF (group 1: 18.8%, group 2: 35.7%, group 3: 60.6%) as the PR interval increased with each stage increment. At pre-TAVI ECG, the duration of the PR interval increased with the staging severity (*p* = 0.02). A PR interval ≥ 240 ms was only seen in groups 2 and 3.

### 3.2. Staging and ECG Outcomes after TAVI

The median duration of hospitalization was 8 days (range 6–13 days). Mean duration of the procedure was 122 min (range 101–140 min). After TAVI, 47% of patients exhibited new conduction disturbances with new LBBB (*n* = 41, 34%) being the most frequent, new RBB in 2.5% (*n* = 3) and HD-AVB in 10% (*n* = 12) (Table 2). Overall, 17 patients (14.7%) required PPMI and 6 patients (5%) died of cardiovascular cause. No patient in group 1 developed HD-AVD whereas new LBBB was frequent in groups 2 and 3. PR interval increased in group 2 (*p* = 0.006) whereas QRS duration increased after TAVI in each group (*p* < 0.0001). Only 1 patient in group 1 had a PR interval ≥ 240 ms and none of them had a QRS ≥ 150 ms. The rate of permanent AF remained similar after TAVI and did not change between groups. However, 21 patients (17.65%) presented with resolutive paroxysmal AF with a higher rate for each group increment (group 1: *n* = 0, 0%; group 2: *n* = 11, 15.7%; group 3: *n* = 10, 30.3%) (*p* = 0.012). Patients in group 3 had the higher rate of PPMI (group 1: *n* = 1, 6.3%; group 2: *n* = 7, 10%; group 3: *n* = 9, 27.3%) (*p* = 0.012). No patient in group 1 died during the hospital stay (Figure 1).

### 3.3. Predictors of PPMI and In-Hospital Mortality

Several factors were associated with an increased risk of PPMI (Table 3): (a) before TAVI: AF as a comorbidity (*p* = 0.023), low pre-TAVI heart rate (*p* = 0.047); (b) during TAVI: sedation with local anesthesia (*p* = 0.028), development of HD-AVB (*p* = 0.0005), use of a size 34 mm prosthesis (*p* = 0.012); (c) after TAVI: a PR interval > 200 ms (*p* = 0.027), a QRS ≥ 150 ms (*p* < 0.0001), a higher QRS delta from baseline ECG (*p* = 0.001), and a new BBB (*p* < 0.01). Factors associated with in-hospital mortality were: a large QRS (*p* = 0.022), a pre-TAVI QRS ≥ 150 ms (*p* = 0.009), a HD-AVB immediately after implantation (*p* = 0.002), and the duration of the procedure (*p* = 0.003).

## 4. Discussion

To the best of our knowledge, this is the first study to specifically evaluate the relationship between ASCDS and ECG changes after TAVI. The main findings in our population are as follows: (a) conduction disorders are frequent after TAVI with new LBBB being the most common, especially in advanced ASCDS stages; (b) patients with ASCDS stage 1 are relatively free of HD-AVD, PPMI and peri-procedural paroxysmal AF, the latter occurring in one-third of patients in group 3; (c) unlike HD-AVB immediately after implantation, the occurrence of new LBBB had no impact on hospital outcomes.

### 4.1. TAVI-Related ECG Changes

New conduction disturbances are relatively frequent after TAVI, some of them with poor prognosis (slow LVEF recovery, conduction dyssynchrony, AF, prolonged hospitalization) [17,18,19]. Most of our patients had an increase in PR interval and QRS duration after TAVI. The incidence of LBBB was 34%, HD-AVB 10%, PPMI 14.7%, and paroxysmal AF 17.7%, which was comparable to large, randomized studies in high-risk patients using the CoreValve system [20,21,22]. Evidence indicates that conduction disturbances associated with TAVI stem from direct or indirect harm to conductive tissue, triggered by interactions with interventional equipment and the stent frame. Pre-existing first-degree AVB or RBBB, severe annular calcifications, lower implant depth, use of a large prosthesis, new LBBB, a PR interval > 200 ms, a QRS ≥ 150 ms and AV block occurring immediately after implantation are all well-known risk factors for PPMI [4,5,23]. In our study, most of these findings were also found to be associated with PPMI. A pre-TAVI QRS ≥ 150 ms and a HD-AVB immediately after implantation predicted in-hospital mortality. On the other hand, the occurrence of a new LBBB had no impact on hospital outcomes. In recent years, numerous studies have investigated the prognostic impact of the occurrence of a new LBBB after TAVI and have shown diverging results. In recent meta-analyses, no significant correlation has been established between the emergence of TAVI-induced LBBB and mortality from all causes [17,18,19,24,25,26].

### 4.2. Cardiac Damage and Post-TAVI Conduction Disorders

In AS, not only the severity of the stenosis but also the extent of cardiac damage has a significant impact on the patient’s prognosis. The more severe the extra-valvular lesions, the more the outcome is compromised with a progressive increase in mortality for each stage increment [14,15]. The prognostic value of this staging system was subsequently tested in populations with severe AS undergoing TAVI, also showing an increase in mortality with the extent of cardiac damage [27,28]. However, to the best of our knowledge, the correlation between ASCDS and ECG changes after TAVI has never been studied. It is known that cardiac electrophysiology can be altered by chronic changes in loading conditions, as in AS [29]. Therefore, with disease progression, the vulnerability of conductive tissues to any mechanical stress, as in the case of TAVI, may increase as extravalvular damage increases. In our population, at pre-TAVI ECG, the duration of the PR interval increased with the staging severity (*p* = 0.02) while there was no significant difference in QRS duration according to ASCDS group. The percentage of bundle branch block at baseline was low, with no significant difference between groups. We reported that the likelihood of conduction disturbances differed depending on the extent of ASCDS prior to TAVI. After TAVI, the increase in PR interval and in QRS duration was more pronounced in advanced ASCDS stages. Only 1 patient in group 1 had a PR interval ≥ 240 ms and none of them had a QRS ≥ 150 ms. The incidence of new LBBB and HD-AVB increased progressively for each stage increment. As a result, the rate of PPMI was the highest in advanced ASCDS stages with 27.3% in group 3, 10% in group 2 compared to 6% in group 1. In a retrospective study assessing prognostic impact of ASCDS in 262 patients, the rate of PPMI did not differ between groups 0/1, 2, 3 and 4 [27]. In this study, the distribution of patients within each ASCDS category differed from our study and the staging criteria used were slightly different. These factors may explain the differences observed and underline the importance of a consensus on a unique ASCDS system for future studies. Although rhythm type did not change significantly after TAVI, the rate of paroxysmal AF increased with higher ASCDS stage. No patient in group 1 developed paroxysmal AF. Taken together, these findings suggest that the increased magnitude of cardiac damage pre-TAVI corresponds to an elevated probability of post-procedural tissue conduction disturbances. Thus, ASCDS appears to be effective in identifying a population at low risk for arrhythmias or PPMI after TAVI. These results could help (1) identify patients who could benefit from the use of the pacing over the wire technique or in whom rapid removal of the right ventricular pacing lead after the procedure could be performed, (2) monitor preferentially stage 1 patients in a cardiology room rather than in an intensive care unit.

### 4.3. Study Limitations

Our study has several limitations. First, its monocentric and retrospective nature can necessarily lead to a selection bias. Second, the total number of patients examined was relatively low to provide sufficient statistical power. Third, the distribution of the population in each stage was not uniform, with a low number of patients in initial stages 0–1 or 3–4, which is why they were logically grouped in stage 1 or 3, corresponding to left chamber cardiac involvement or upstream repercussions of AS on the pulmonary circulation and the right ventricle. This limits the precise evaluation of the prognostic impact of the original ASCDS classification, but nevertheless constitutes an initial approach. Fourth, ASCDS is based on echocardiographic parameters which may be subject to measurement errors and intra- and inter-operator variability. Finally, it should be noted that our study only concerned self-expandable prostheses and it remains to be determined whether our results could also be transposed to balloon expandable prostheses. Future studies addressing those issues are needed to confirm and generalize these results.

## 5. Conclusions

Despite technological advancements, increased operator experience and improvements in terms of efficiency and safety of the TAVI procedure, the incidence of peri- and post-procedure conduction abnormalities remains high. ASCDS shows potential for identifying patients at risk of PPMI requirement and patients at low risk of conduction disorders. It could therefore help practitioners to define patient monitoring strategy after TAVI.

Future prospective studies in larger and more diverse cohort of patients are needed to confirm and generalize these results.

## Figures and Tables

**Figure 1 jcm-13-00409-f001:**
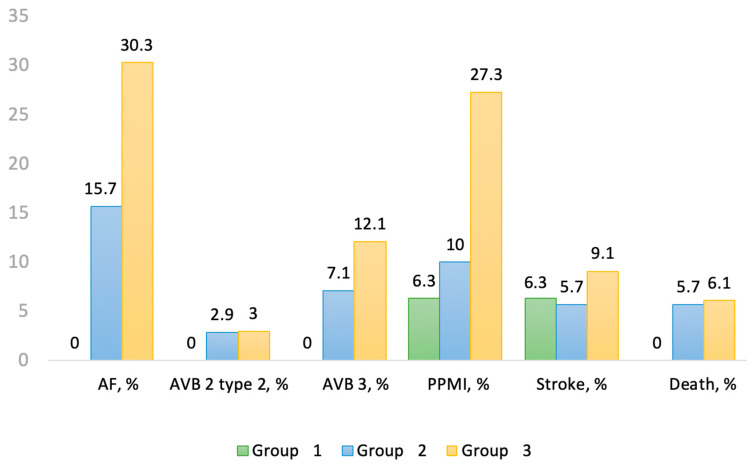
Outcomes after TAVI between the different groups. AF: atrial fibrillation, AVB: atrioventricular block, PPMI: permanent pacemaker implantation.

**Table 1 jcm-13-00409-t001:** Baseline characteristics of the study population before TAVI.

	Total	Group 1	Group 2	Group 3	*p*-Value
Clinical characteristics					
Number of patients (%)	119 (100%)	16 (13.5%)	70 (58.8%)	33 (27.7%)	
Age (years)	84 (79–86)	78 (73.5–84.5)	83.5 (79–87)	85 (81–87)	0.017
Male, (%)	64 (53.78)	9 (56.3)	37 (52.9)	18 (54.6)	0.97
Body mass index, kg/m^2^	25.51 (23.2–29.6)	27.1 (23.3–29.5)	26.4 (23.1–30.1)	24.9 (23.5–26.9)	0.58
Current smoker (%)	45 (37.8)	8 (50)	26 (37.1)	11 (33.3)	0.30
Hypertension n (%)	106 (89.1)	14 (87.5)	62 (88.6)	30 (90.9)	0.82
Diabetes (%)	35 (29.4)	6 (37.5)	20 (28.6)	9 (27.3)	0.52
Dyslipidemia (%)	96 (80.7)	14 (87.5)	55 (78.6)	27 (81.8)	0.86
Atrial fibrillation (%)	48 (40.3)	3 (18.8)	25 (35.7)	20 (60.6)	0.003
Stroke (%)	22 (18.5)	0 (0)	15 (21.4)	7 (21.2)	0.19
Prior peripheral vascular disease (%)	59 (49.6)	7 (43.8)	37 (52.9)	15 (45.5)	0.90
Coronary artery disease (%)	53 (44.5)	8 (50)	29 (41.4)	16 (48.5)	0.89
Chronic kidney diseases (eGFR < 45 mL/min)	50 (42.0)	5 (31.3)	29 (41.4)	16 (48.5)	0.25
Medical therapy					
Beta-blockers (%)	84 (70.6)	9 (56.3)	49 (70)	26 (78.8)	0.11
Calcium antagonists (%)	28 (23.5)	4 (25)	16 (22.9)	8 (24.2)	1.00
Amiodarone (%)	21 (17.7)	2 (12.5)	12 (17.1)	7 (21.2)	0.57
Procedure					
Euroscore II (%)	3.98 (2.51–6.02)	3.92 (2.8–4.9)	3.64 (2.35–6.06)	4.8 (2.8–11.27)	0.24
Calcium score	2471.45 (1652–3768)	1582 (869–3197)	2990 (1853–4111)	2211 (1855–3117)	0.10
Pre-dilatation (%)	38 (31.9)	3 (18.8)	23 (32.9)	12 (36.4)	0.28
Post-dilatation (%)	15 (12.6)	1 (6.3)	12 (17.1)	2 (6.1)	0.66
Prosthesis size					0.55
23 mm (%)	8 (6.72)	0 (0)	4 (5.7)	4 (12.1)	
26 mm (%)	33 (27.7)	5 (31.3)	19 (27.1)	9 (27.3)	
29 mm (%)	45 (37.8)	7 (43.8)	27 (38.6)	11 (33.3)	
34 mm (%)	33 (27.7)	4 (25)	20 (28.6)	9 (27.3)	
Approach (%)					0.32
Femoral	103 (86.6)	14 (87.5)	60 (85.7)	29 (87.9)	
Trans-Axillary	15 (12.6)	2 (12.5)	10 (14.3)	3 (9.1)	
Carotid	1 (0.84)	0 (0)	0 (0)	1 (3)	
General anesthesia (%)					0.42
General	80 (67.2)	7 (43.8)	52 (74.3)	21 (63.6)	
Local anesthesia + sedation	39 (32.8)	9 (56.3)	18 (25.7)	12 (36.4)	
Procedure time(min)	122 (101–140)	123.5 (102.5–132.5)	123 (102–140)	115 (99–147)	0.84
Electrocardiogram					
Heart rate (bpm)	71 (62–81)	73.5 (65–85)	69 (60–80)	72 (62–81)	0.59
Rhythm					<0.0001
Sinus (%)	94 (78.99)	16 (100)	61 (87.1)	17 (51.5)	
Atrial fibrillation (%)	25 (21.01)	0 (0)	9 (12.9)	16 (48.5)	
PR interval (ms)	180 (164–204)	165 (161–177)	180 (164–204)	202 (180–214)	0.020
<200 ms (%)	63 (67.0)	15 (93.8)	40 (57.1)	8 (24.2)	
200–239 ms (%)	20 (21.3)	1 (6.3)	13 (18.6)	6 (18.2)	
≥240 ms (%)	11 (11.70)	0 (0)	8 (11.4)	3 (9.1)	
QRS (ms)	96 (84–112)	89 (78–98)	96 (86–119)	94 (84–110)	0.13
<120 ms (%)	96 (80.7)	14 (87.5)	54 (77.1)	28 (84.9)	
120–149 ms (%)	17 (14.3)	2 (12.5)	10 (14.3)	3 (15.1)	
≥150 ms (%)	6 (5.04)	0 (0)	6 (8.6)	0 (0)	
Prior bundle branch block					0.47
Left (%)	10 (8.4)	1 (6.3)	7 (10)	2 (6.1)	
Right (%)	6 (5.04)	1 (6.3)	5 (7.1)	0 (0)	

**Table 2 jcm-13-00409-t002:** ECG outcomes after TAVI.

	Total	Group 1 *n* = 16 (13.5%)	Group 2 *n* = 70 (58.8%)	Group 3 *n* = 33 (27.7%)	*p*-Value
Electrocardiogram					
Heart rate (bpm)	73 (65–86)	72 (61–95)	72 (65–81)	79 (67.5–88.5)	0.41
Rhythm					
Atrial fibrillation (%)	25 (21.7)	0 (0)	9 (12.9)	16 (48.5)	<0.0001
PR interval (ms)	190 (166–214)	170 (162–194)	192 (172–215)	214 (168–226)	0.052
<200 ms (%)	55 (63.2)	13 (81.3)	36 (51.4)	6 (18.2)	
200–239 ms (%)	19 (21.8)	2 (12.5)	11 (15.7)	3 (18.2)	
≥240 ms (%)	13 (14.9)	1 (6.3)	9 (12.9)	3 (9.1)	
Delta PR	8 (−6–20)	4 (−3–23) *	8 (−6–18) *	2 (−22–26)	0.71
QRS (ms)	122 (96–140)	101 (88–129)	122 (94–140)	124.5 (103–142)	0.13
<120 ms (%)	54 (47.4)	10 (62.5)	30 (42.9)	14 (42.4)	
120–149 ms (%)	42 (36.8)	6 (37.5)	23 (32.9)	13 (39.4)	
≥150 ms (%)	18 (15.8)	0 (0)	13 (18.6)	5 (15.2)	
Delta QRS	8 (2–36)	11 (6–24) *	5 (0–33) *	19 (4–54) *	0.082
New bundle branch block					0.17
Left (%)	41 (34.5)	4 (25)	24 (34.3)	13 (39.4)	
Right (%)	3 (2.52)	0 (0)	1 (1.4)	2 (6.1)	
Complications					
Paroxysmal atrial fibrillation (%)	21 (17.7)	0 (0)	11 (15.7)	10 (30.3)	0.012
High degree AV block (%)	12 (10)	0 (0)	7 (10)	5 (15)	0.38
Permanent pacemaker implantation (%)	17 (14.3)	1 (6.3)	7 (10)	9 (27.3)	0.022
Stroke (%)	8 (6.7)	1 (6.3)	4 (5.7)	3 (9.1)	0.32
Death (%)	6 (5)	0 (0)	4 (5.7)	2 (6.1)	0.52

* *p* < 0.05, difference between post and pre-TAVI.

**Table 3 jcm-13-00409-t003:** Variables associated with PPMI.

Variables	OR (95% CI)	*p*-Value	Units
Local vs. general anesthesia	2.77 (1.11–6.89)	0.028	
Heart rate ECG pre-TAVI	0.96 (0.93–1)	0.047	1 beat
Duration QRS ECG pre-TAVI	1.16 (1–1.34)	0.051	10 ms
PR pre-TAVI 200–239 ms vs. <200 ms	3.35 (0.97–11.59)	0.056	
Per-procedure: AV block 2 type 2 vs. No AV block	142.6 (8.71–2333.5)	0.0005	
Per-procedure: AV block 3 vs. No AV block	6.2 (2.19–17.54)	0.0006	
Prosthesis size 34 vs. 29	5.27 (1.45–19.17)	0.012	
Atrial fibrillation	2.95 (1.16–7.51)	0.023	
Heart rate ECG post-TAVI	0.97 (0.94–1)	0.082	1 beat
Interval PR ECG post-TAVI	1.20 (1.06–1.36)	0.004	10 ms
Delta PR	1.02 (1–1.04)	0.11	1 ms
Duration QRS ECG post-TAVI	1.55 (1.26–1.91)	<0.0001	10 ms
Delta QRS	1.03 (1.01–1.05)	0.001	1 ms
ECG post-TAVI: left bundle-branch block vs. No bundle-branch block	5.76 (1.62–20.44)	0.007	
ECG post-TAVI: right bundle-branch block vs. No bundle-branch block	7.15 (1.44–35.49)	0.016	
PR post-TAVI < 200 ms vs. ≥240 ms	0.13 (0.02–0.79)	0.027	
PR post-TAVI 200–239 ms vs. <200 ms	6.59 (1.2–35.99)	0.030	
QRS post-TAVI ≥ 150 ms vs. <120 ms	16.97 (4.55–63.29)	<0.0001	
QRS post-TAVI 120–149 ms vs. ≥150 ms	0.17 (0.06–0.48)	0.0009	
Post-TAVI: AV block 3 vs. No AV block	9.95 (3.67–26.92)	<0.0001	

## Data Availability

The data presented in this study are available on request from cecile.wegria@chuliege.be.

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
