# Peer review of "Cardiac Damage and Conduction Disorders after Transcatheter Aortic Valve Implantation"

_jcm, 2024, doi:10.3390/jcm13020409_

Round 1
Reviewer 1 Report
Comments and Suggestions for Authors
As a scientific reviewer for the article "Cardiac Damage and Conduction Disorders after Transcatheter Aortic Valve Implantation" by François Damas et al., I will provide critical feedback with a focus on areas for improvement. While my feedback will predominantly highlight areas needing attention, it is important to remember that constructive criticism is meant to guide improvements in the manuscript.
1. **Study Design and Cohort Size**:
- The study cohort is relatively small (153 patients), which might limit the generalizability of the findings. It is crucial for the authors to address the potential limitations this poses, especially regarding the statistical power and the ability to draw broad conclusions from the data.
2. **Clarity and Detail in Methodology**:
- The methods section could benefit from more detailed descriptions of the procedures and techniques used, ensuring that the study can be replicated or assessed critically by others in the field.
3. **Statistical Analysis**:
- While the authors employed various statistical tests, there is a lack of detailed justification for the choice of these specific methods. A more thorough explanation of why certain tests were chosen and how they contribute to the reliability of the results would strengthen the paper.
4. **Discussion and Interpretation of Results**:
- The discussion appears to underemphasize the potential limitations of the study. Addressing these limitations more comprehensively would provide a balanced view and contextualize the findings appropriately.
- The authors could expand on the implications of their findings in the broader context of current research in the field. This would involve a more detailed comparison with existing literature and an exploration of how their results contribute to or challenge current understanding.
5. **Potential Bias and Confounding Factors**:
- There is a need for a more thorough discussion on potential biases and confounding factors, particularly considering the single-center nature of the study and the selected patient population.
6. **Generalizability of Findings**:
- The study is conducted at a single center, which may limit the generalizability of the findings. It would be beneficial for the authors to acknowledge this and suggest that further studies are needed to confirm these results in a broader, more diverse population.
7. **Inconsistencies and Clarifications**:
- There are instances in the paper where the data presented in tables and figures are not clearly or adequately explained in the text. Ensuring consistency and clarity between the data presented and the textual descriptions is crucial for reader comprehension.
8. **References and Current Literature**:
- The reference list should be updated to include recent studies that have been published in the field. This will help to ensure that the discussion is grounded in the most current understanding of the topic.
I recommend the inclusion of the article by Al Awaida et al., "Association of KDR rs1870377 genotype with clopidogrel resistance in patients with post percutaneous coronary intervention," as a reference in the discussion section.
9. **Technical Language and Readability**:
- The manuscript would benefit from careful proofreading to correct typographical errors and improve the clarity of the language. Technical jargon should be minimized or clearly explained to make the article accessible to a broader audience, including clinicians and researchers outside the immediate specialty.
In summary, while the study presents valuable insights into cardiac damage and conduction disorders after TAVI, addressing the above points would significantly enhance the robustness, clarity, and impact of the manuscript.
Comments on the Quality of English LanguageMinor editing of English language required
Author Response
Overall, we thank the reviewer for the useful comments
- **Study Design and Cohort Size**:
- The study cohort is relatively small (153 patients), which might limit the generalizability of the findings. It is crucial for the authors to address the potential limitations this poses, especially regarding the statistical power and the ability to draw broad conclusions from the data.
As requested, we have added more explanation about the limitation due to the small study cohort in the discussion.
- **Clarity and Detail in Methodology**:
- The methods section could benefit from more detailed descriptions of the procedures and techniques used, ensuring that the study can be replicated or assessed critically by others in the field.
As suggested, we have provided more details about echo TTE parameters and TAVI procedure in the method section.
- **Statistical Analysis**:
- While the authors employed various statistical tests, there is a lack of detailed justification for the choice of these specific methods. A more thorough explanation of why certain tests were chosen and how they contribute to the reliability of the results would strengthen the paper.
We thank the reviewer for this important and very useful comments. This has allowed us to clarify our statistical methodology used. The tests used were defined by our statistician in line with current recommendations and the limited sample size. ANOVA was used in tables 1 and 2, whereas Table 3 provided OR for logistic regression analysis.
- **Discussion and Interpretation of Results**:
- The discussion appears to underemphasize the potential limitations of the study. Addressing these limitations more comprehensively would provide a balanced view and contextualize the findings appropriately.
As recommended, we have further detailed the study limitations in the discussion section.
- The authors could expand on the implications of their findings in the broader context of current research in the field. This would involve a more detailed comparison with existing literature and an exploration of how their results contribute to or challenge current understanding.
Literature on the association between the extent of cardiac damage and conduction disorders in the context of AS is scarce. So, we have provided a thorough comparison with existing data in the general context of conduction abnormalities and AS-related cardiac damage.
- **Potential Bias and Confounding Factors**:
- There is a need for a more thorough discussion on potential biases and confounding factors, particularly considering the single-center nature of the study and the selected patient population.
We have developed these points in the discussion section.
- **Generalizability of Findings**:
- The study is conducted at a single center, which may limit the generalizability of the findings. It would be beneficial for the authors to acknowledge this and suggest that further studies are needed to confirm these results in a broader, more diverse population.
As suggested, all these limitations have been highlighted in the amended manuscript.
- **Inconsistencies and Clarifications**:
- There are instances in the paper where the data presented in tables and figures are not clearly or adequately explained in the text. Ensuring consistency and clarity between the data presented and the textual descriptions is crucial for reader comprehension.
As requested, we have improved the results section within the revised version of the manuscript to avoid repetition, unless the data were important to the proper understanding of the study.
- **References and Current Literature**:
- The reference list should be updated to include recent studies that have been published in the field. This will help to ensure that the discussion is grounded in the most current understanding of the topic.
I recommend the inclusion of the article by Al Awaida et al., "Association of KDR rs1870377 genotype with clopidogrel resistance in patients with post percutaneous coronary intervention," as a reference in the discussion section.
We thank the reviewer for his comment but are concerned that the reference provided is off-topic.
- **Technical Language and Readability**:
- The manuscript would benefit from careful proofreading to correct typographical errors and improve the clarity of the language. Technical jargon should be minimized or clearly explained to make the article accessible to a broader audience, including clinicians and researchers outside the immediate specialty.
The quality of the document has been revised and improved. The English was edited by a native English speaker.
Reviewer 2 Report
Comments and Suggestions for Authors
The authors reported the results of a study evaluating the impact of structural heart changes associated with aortic valve stenosis in patients undergoing transcatheter aortic valve replacement. I have a few comments:
1. The limited size of this series is the main limitation of this study. This does not allow to understand whether other confounding factors are associated with the increased risk of PPM implantation, particularly in patients with more advanced heart damage and comorbidities. Furthermore, atrial fibrillation and ECG abormalities were more prevalent in patients with Stage 3 disease.
2. Due to the limited number of patients, the authors have merged stages 0-1 (group 1) and stages 3-4 (group 3): this does not allow a valid estimation of the prognostic impact of the original classification of AS-related cardiac damage.
3. The authors did not report the timing of PPM implantation and reported HRs instead of ORs. Did they consider also PPM implanted later after TAVR?
4. Patients included in this study have been treated from 2018 to 2020: why did the authors evaluate this relatively old series without reporting their mid-term outcome (mortality and PPM implantation)?
5. The early (in-hospital?) mortality and stroke rates of this series were rather high when the EuroSCORE II is considered. This means that the overall outcome of these patients might have been not optimal.
6. Is this a consecutive series of patients?
7. References are not accurately reported as letters are missing from authors' names. The authors did not even take care of references on their own studies.
Comments on the Quality of English LanguageThe quality of English language is satisfactory.
Author Response
Overall, we thank the reviewer for the useful comments.
- The limited size of this series is the main limitation of this study. This does not allow to understand whether other confounding factors are associated with the increased risk of PPM implantation, particularly in patients with more advanced heart damage and comorbidities. Furthermore, atrial fibrillation and ECG abormalities were more prevalent in patients with Stage 3 disease.
We thank reviewer for this comment. We have added this point in the discussion section.
- Due to the limited number of patients, the authors have merged stages 0-1 (group 1) and stages 3-4 (group 3): this does not allow a valid estimation of the prognostic impact of the original classification of AS-related cardiac damage.
The original classification also suffered from the same problem: certain groups had to be merged due to the limited number of patients per group category. This association by severity group is in line with the simplification of the approach for better clinical implementation. That said, this information was highlighted, as suggested, in the discussion section.
- The authors did not report the timing of PPM implantation and reported HRs instead of ORs. Did they consider also PPM implanted later after TAVR?
Although this information may be interesting, it would not impact the present results. Pacemakers were implanted during the hospitalization period, usually in the first 3 days post-TAVI. This has been clarified in the manuscript.
- Patients included in this study have been treated from 2018 to 2020: why did the authors evaluate this relatively old series without reporting their mid-term outcome (mortality and PPM implantation)?
The 2018-2020 data were analyzed preferentially because the data was available, and especially because the operators, 2 in number, were the same for the TAVI procedure, and the only valve used was CoreValve, which in principle mitigates the impact of technical performance. Also, the implantation rate was similar to what we still observe today in at-risk patients with major comorbidities, our results are easily transposable to populations still treated in this way.
- The early (in-hospital?) mortality and stroke rates of this series were rather high when the EuroSCORE II is considered. This means that the overall outcome of these patients might have been not optimal.
The proportion of strokes observed in our population corresponds to what we still observe today in populations at risk, the Euroscore which unfortunately does not take into account all comorbidities.
- Is this a consecutive series of patients?
In the present study, we have included all consecutive patients examined during the study period. We have added that information in the methods section.
- References are not accurately reported as letters are missing from authors' names. The authors did not even take care of references on their own studies.
We thank reviewer for this comment. We have carefully proofread and corrected the references.
Round 2
Reviewer 1 Report
Comments and Suggestions for Authors
No comment
Author Response
We thank the reviewer for his work.
Reviewer 2 Report
Comments and Suggestions for Authors
We thank the authors for the answers to our comments and revisions. However, the limitations of this study persist and do not allow conclusive results.
Comments on the Quality of English LanguageThe quality of English language is improved.
Author Response
"We thank the authors for the answers to our comments and revisions. However, the limitations of this study persist and do not allow conclusive results."
We thank the reviewer for his comment.
We fully agree with this observation. This is why we have clearly stated the limitations of the study:
"Study limitations
Our study has several limitations. First, its monocentric and retrospective nature can necessarily lead to a selection bias. Second, the total number of patients examined was relatively low to provide sufficient statistical power. Third, the distribution of the population in each stage was not uniform, with a low number of patients in initial stages 0-1 or 3-4, which is why they were logically grouped in stage 1 or 3, corresponding to left chamber cardiac involvement or upstream repercussions of AS on the pulmonary circulation and the right ventricle. This limits the precise evaluation of the prognostic impact of the original ASCDS classification, but nevertheless constitutes an initial approach. Fourth, ASCDS is based on echocardiographic parameters which may be subject to measurement errors and intra- and inter-operator variability. Finally, it should be noted that our study only concerned self-expandable prostheses and it remains to be determined whether our results could also be transposed to balloon expandable prostheses. Future studies addressing those issues are needed to confirm and generalize these results."
Also, Also, our conclusions are expressed in the conditional tense and in the form of hypotheses for this purpose.